# Characterization of an *Agarophyton chilense* Oleoresin Containing PPARγ Natural Ligands with Insulin-Sensitizing Effects in a C57Bl/6J Mouse Model of Diet-Induced Obesity and Antioxidant Activity in *Caenorhabditis elegans*

**DOI:** 10.3390/nu13061828

**Published:** 2021-05-27

**Authors:** Claudio Pinto, María Raquel Ibáñez, Gloria Loyola, Luisa León, Yasmin Salvatore, Carla González, Víctor Barraza, Francisco Castañeda, Rebeca Aldunate, Loretto Contreras-Porcia, Karen Fuenzalida, Francisca C. Bronfman

**Affiliations:** 1Postgraduate Department, Faculty of Veterinary Sciences, Universidad Austral de Chile, Valdivia 5110566, Chile; claudiopintovet@gmail.com; 2Center for Aging and Regeneration (CARE), Department of Cellular and Molecular Biology, Faculty of Biological Sciences, Pontificia Universidad Católica de Chile, Santiago 8320000, Chile; 3Department of Physiology, Faculty of Biological Sciences, Pontificia Universidad Católica de Chile, Santiago 8320000, Chile; raquel.ibanez@gmail.com (M.R.I.); galoyola@gmail.com (G.L.); mleon@bio.puc.cl (L.L.); ysalvatore@uc.cl (Y.S.); cdgonzalez@bio.puc.cl (C.G.); vbarraza@bio.puc.cl (V.B.); 4Institute of Biomedical Sciences (ICB), Faculty of Medicine, Universidad Andres Bello, Santiago 8320000, Chile; 5Department of Ecology and Biodiversity, Faculty of Life Sciences, Universidad Andres Bello, Santiago 8320000, Chile; fra.castaneda@gmail.com (F.C.); lorettocontreras@unab.cl (L.C.-P.); 6Quintay Marine Research Center (CIMARQ), Faculty of Life Sciences, Universidad Andres Bello, Valparaiso, Quintay 2480000, Chile; 7Center of Applied Ecology and Sustainability (CAPES), Santiago 8331150, Chile; 8Instituto Milenio en Socio-Ecología Costera (SECOS), Santiago 8370251, Chile; 9Faculty of Sciences, School of Biotechnology, Universidad Santo Tomas, Santiago 8320000, Chile; raldunate@santotomas.cl

**Keywords:** natural lipids, seaweeds, *Agarophyton chilense*, PPARγ, insulin resistance, obesity, antioxidants, *Caenorhabditis elegans*, nutraceuticals, Gracilex*^®^*

## Abstract

The biomedical potential of the edible red seaweed *Agarophyton chilense* (formerly *Gracilaria chilensis*) has not been explored. Red seaweeds are enriched in polyunsaturated fatty acids and eicosanoids, which are known natural ligands of the PPARγ nuclear receptor. PPARγ is the molecular target of thiazolidinediones (TZDs), drugs used as insulin sensitizers to treat type 2 diabetes mellitus. Medical use of TZDs is limited due to undesired side effects, a problem that has triggered the search for selective PPARγ modulators (SPPARMs) without the TZD side effects. We produced *Agarophyton chilense* oleoresin (Gracilex*^®^*), which induces PPARγ activation without inducing adipocyte differentiation, similar to SPPARMs. In a diet-induced obesity model of male mice, we showed that treatment with Gracilex*^®^* improves insulin sensitivity by normalizing altered glucose and insulin parameters. Gracilex*^®^* is enriched in palmitic acid, arachidonic acid, oleic acid, and lipophilic antioxidants such as tocopherols and β-carotene. Accordingly, Gracilex*^®^* possesses antioxidant activity in vitro and increased antioxidant capacity in vivo in *Caenorhabditis elegans.* These findings support the idea that Gracilex^®^ represents a good source of natural PPARγ ligands and antioxidants with the potential to mitigate metabolic disorders. Thus, its nutraceutical value in humans warrants further investigation.

## 1. Introduction

The edible Chilean red macroalgae *Agarophyton chilense* (formerly *Gracilaria chilensis*) [1], commonly known as “pelillo”, has been used as a food and medicinal herb since pre-Hispanic times in Chile, as indicated by findings at the archaeological site of Monte Verde (~14,000 years ago) [2]. *Agarophyton chilense (A. chilense)* is distributed in New Zealand (including Chatham Island) and South America [3,4,5,6], and although this species has good nutritional potential, it is mainly exploited for the extraction of agar-agar, a hydrocolloid [7,8].

The reported bioactivity of seaweed is varied and includes antidiabetic, anti-inflammatory, antioxidant, and anti-neurodegenerative activity [9,10]. Thus, in recent years, there has been increased interest in identifying novel bioactive compounds that demonstrate health benefits to confirm and increase the added value of edible red seaweeds [8,11].

Red algae such as *A. chilense* are an excellent source of healthy essential fatty acids such as polyunsaturated fatty acids (PUFAs), including eicosapentaenoic acid, arachidonic acid, and oleic, linoleic, and α-linolenic fatty acids. Particularly in the *Gracilaria* genus, lipid extraction has revealed a complex lipid composition of glycerolipids and omega-3 and omega-6 PUFAs and their oxidized derivatives, known as oxylipins [12,13,14,15]. Oxylipins are synthesized either by the metabolic action of specific lipoxygenases (LOXs) or by nonenzymatic reactions on PUFAs [16,17,18]. Interestingly, the oxylipin structure is similar or equivalent to that of mammalian prostaglandins, leukotrienes, and eicosanoids [19], which are well-known ligands of peroxisome proliferator-activated receptors (PPARs) [20,21,22].

The PPAR nuclear receptor subfamily comprises three receptor subtypes (PPARα, PPARß/∂, and PPARγ) encoded by distinct genes, and they are obligate heterodimers of the RXR receptors [23]. PPARs have attracted increased attention as therapeutic targets given the physiological relevance of the processes in which they are involved, including lipid and glucose homeostasis, and because they have anti-inflammatory and anti-oxidant activity [23,24,25,26]. The three PPARs have natural ligands, including PUFAs and eicosanoids [27,28,29,30,31]. All of these lipids activate PPARs with low affinity (micromolar range) compared to high-affinity synthetic agonists such as fibrates and thiazolidinediones (TZDs). In particular, fibrates, which are drugs used to treat dyslipidemia, target PPARα [26,32] while TZDs act as full agonists of PPARγ. TZDs were designed in the late 1990s for the treatment of type 2 diabetes mellitus (T2DM), to regulate blood sugar levels and improve insulin sensitivity in patients with, and animal models of, T2DM [33]. However, in early 2007, the adverse effects of these drugs, including weight gain, fluid retention, osteoporosis, and heart failure, among others, were described [33,34,35]. These side effects have led to restricted access in the United States and a recommendation for removal from the market in Europe and other jurisdictions [36]. At the molecular level, TZDs, such as rosiglitazone, stabilize the ligand-binding site of PPARγ in an “on” or “off” state. In contrast, selective PPARγ modulators (SPPARMs) generate various dynamic and slow conformational changes by recruiting different coactivators and corepressors compared to TZDs, thus inducing a different gene expression profile and reduced side effects compared to TZDs [37].

T2DM is a chronic endocrine disease characterized by hyperglycemia in the blood and resistance to the action of insulin, leading to severe neurological and cardiovascular lesions. Although there are genetic risk factors, obesity is considered the explaination for 60–80% of diabetes cases [38,39,40].

There are various classes of oral drugs for glycemic management, such as metformin, sulfonylureas, and TZDs. Currently, the first-line oral antihyperglycemic is metformin. However, the majority of patients with T2DM show a deterioration in blood glucose levels over time that cannot be controlled with metformin alone; consequently, a second-line drug to maintain glycemic control is included in the treatment [41,42]. Recently, the use of nutraceuticals, such as berberine extracts and omega-3 fatty acids, was suggested for the management of glycemia and dyslipidemia associated with T2DM [43]. Experts have emphasized the use of nutraceuticals and functional foods to treat or mitigate T2DM based on the advantages of using a mix of molecules with simultaneous effects. The Asian continent is familiar with medicine derived from natural products. Therefore, interest in the use of nutraceuticals, including products derived from the sea such as seaweeds [44], for the treatment of T2DM has increased [45].

Despite the rich diversity of lipids and the unique phytochemical composition of red macroalgae lipids, the biomedical potential of lipids derived from *A. chilense* has not been addressed. The principal objective of our work was to study the biomedical potential of *A. chilense* oleoresin by evaluating its capacity to activate PPARγ and modulate the metabolic dysfunction induced by obesity. We also characterized some of the lipids contained in the oleoresin, such as fatty acids, tocopherols, and β-carotene, and the antioxidant properties in vitro and in vivo using *Caenorhabditis elegans (C. elegans)* as a model. The oleoresin was produced by organic solvent extraction of dried algae following the procedure published in the patent application WO/2014/186913, and its trademark registered name is Gracilex*^®^.* Our results indicate that Gracilex*^®^* contains PPARγ activators acting as partial agonists, since they do not induce adipocyte differentiation like the TZD rosiglitazone. Consistently, treatment of male mice fed a high-fat diet (HFD) with Gracilex*^®^* reduced glucose and insulin levels increased by the HFD. Moreover, Gracilex*^®^* has an antioxidant capacity correlating with high levels of tocopherols and β-carotene. Our study is the first to examine whether lipids derived from *A. chilense* have nutraceutical or biomedical applications, emphasizing the idea that *A. chilense* can be used as a source of other molecules in addition to agar-agar.

## 2. Material and Methods

### 2.1. Reagents

HPLC-grade dichloromethane, cyclohexane, and water were purchased from Merck (Darmstad, Germany). Cell culture reagents, namely Dulbecco’s Modified Eagle Medium (DMEM), RPMI 1640, opti-PRO™, GlutaMAX^TM^ I supplement, Tripsin-EDTA 0.5%, and penicillin/streptomycin, were purchased from Gibco BRL (Life Technologies, Inc., Gaithersburg, MD, USA). Fetal bovine serum and horse serum (HS) were obtained from HyClone Laboratories, Inc. (Thermo Fisher Scientific, Waltham, MA, USA) and Lipofectamine^TM^ 2000 from Gibco BRL (Life Technologies, Waltham, MA, USA). Drugs and chemicals were purchased from Sigma Chemical Co. (St. Louis, MO, USA), including FMOC-Leu, 3-(4,5-dimethylthiazol-2-yl)-2,5-diphenyl tetrazolium bromide (MTT), ortho-nitrophenyl-β-galactoside (ONPG), N-acetylcysteine (NAC), cell culture grade dimethyl sulfoxide (DMSO), NP-40, Oil red O, dexamethasone, isobutylmethylxanthine (IBMX), and insulin. Rosiglitazone and T0070907 were purchased from Cayman Chemicals (Ann Arbor, MI, USA) and INT131 was obtained from MolPort (Beacon, NY, USA). FMOC-Leu and INT131 are SPPARMs and rosiglitazone and T0070907 are full-agonist and antagonist of the PPARγ, respectively.

Plasmids used to measure the transcriptional activity of PPARγ (see below) were donated by R.M. Evans’s laboratory (Howard Hughes Medical Institute, Gene Expression Laboratory, The Salk Institute for Biological Studies, La Jolla, CA, USA).

### 2.2. Sampling of Biomass

Vegetative *A. chilense* individuals, approximately 60 cm in length, were collected from two areas along the Chilean coast in the southern zone: Niebla (39.87° S 73.40° W, Los Lagos Region) and Coliumo (36.55° S 72.95° W, Biobío Region). The total biomass (20–30 kg of fresh tissue) was transported to the laboratory at 4 °C and washed with fresh water posteriorly with 1 µm filtered seawater and cleaned before freezing.

### 2.3. Oleoresin Analysis

#### 2.3.1. A. Chilense (Gracilex^®^) Oleoresin Production

Harvested seaweed was frozen at −20 °C for two weeks and then chopped into small pieces (3–5 mm in length) with a knife for the production of oxylipins induced by algal tissue after mechanical damage [16]. Chopped algae were freeze-dried through lyophilization. Then, to produce a lyophilized powder, the dried material was ground using a coffee grinding machine. Dichloromethane was used as an organic solvent for lipid extraction of pulverized dried seaweed. In brief, 50 g of pulverized dry alga was placed in an Erlenmeyer flask with 150 mL of dichloromethane, shaken for 30 min, and filtered. The remaining sediment was extracted once again as indicated above. The whole filtered extract was evaporated, suspended in cyclohexane, and lyophilized to finally obtain an oleoresin of *A. chilense*, whose trademark registered name is Gracilex^®^. The detailed protocol is published in the patent application WO/2014/186913 [46].

#### 2.3.2. Determination of Total Antioxidant Capacity of Gracilex^®^ Using a Cupric Ion Reducing Antioxidant Capacity (CUPRAC) Assay

Oleoresins obtained from dichloromethane extraction of lyophilized spirulina (AquaSolar™) and maqui (Isla Natura™) were prepared as described above for Gracilex^®^ production. The oleoresins were resuspended in DMSO, and 50 µL of sample was taken to measure the antioxidant capacity using an OxiSelect™ TAC Assay Kit according to the manufacturer’s instructions (Cell Biolabs, Inc., San Diego, CA, USA), which is based on the reduction of copper (II) to copper (I), also known as a cupric ion reduction antioxidant capacity (CUPRAC) assay [47]. Uric acid was used as a standard to calculate the milli-equivalents of antioxidant.

#### 2.3.3. Analysis of Lipid and Antioxidant Content of Gracilex^®^

Six preparations of Gracilex^®^ were used to measure tocopherols (α-, γ-, and δ-tocopherol) and β-carotene by HPLC-electrochemical methods. The fatty acid profile of Gracilex^®^ was measured as fatty acid methyl esters by GC-FID. Both analyses were performed at the Center of Molecular Nutrition and Chronic Disease of Pontificia Universidad Católica de Chile according to the Chilean normative procedure NCh2550-ISO5508.

### 2.4. Cellular Studies

#### 2.4.1. Cell Lines

HeLa (ATCC^®^ CCL-2^™^), PC12 (ATCC^®^ CRL-1721^™^), and 3T3-L1 (ATCC^®^ CL-173^™^) cell lines (ATCC, Manassas, VA, USA) were used for our studies. All cells were cultured, following ATTC’s recommendations, in a 5% CO_2_ incubator at 37 °C and 95% humidity.

PC12 cells were grown and cultured in RPMI 1640 medium supplemented with 10% horse serum (HS), 5% fetal bovine serum (FBS), and penicillin/streptomycin (P/S). HeLa and 3T3-L1 cells were grown and cultured in high-glucose DMEM with 10% and P/S. The 3T3-L1 cells were grown under low passages at a maximum of 70% confluence.

#### 2.4.2. Cellular Transfection

A PPAR reporter activity assay was performed by transient transfection of PC12 and HeLa cells with a reporter plasmid containing three tandem repeats of the peroxisomal proliferator response element (PPRE) fused to the herpesvirus thymidine kinase promoter upstream of the coding sequence for luciferase (PPRE-tk-LUC) and a pCMX vector containing full-length murine PPARγ1 [48]. Cells were grown until 60–70% confluence in 24-well collagen-coated plates and transfected with 2 μL of Lipofectamine^TM^ 2000 (Thermo Fisher, Waltham, MA, USA) and 0.3 μg of PPRE-tk-LUC vector, 0.26 μg of pCMX-PPARγ vector, and 0.1 μg of β-galactosidase expression plasmid (CMV-β-Gal) vector (Clontech, now Takara Bio USA Inc., Mountain View, CA, USA) for normalization.

A PPARγ-GAL4 transactivation assay was performed in PC12 cells seeded in 48-well plates at a density of 1 × 10^5^ cells/well in 250 µL of RPMI 1640 medium (10% HS, 5% SFB, and P/S) grown for 24 h and transfected with Lipofectamine^TM^ 2000 (Thermo Fisher, Waltham, MA, USA) with 0.8 µg of reporter plasmid (Gal4-dependent MH100tk-Luc), 0.8 µg of fusion protein plasmid (PPARγGAL4), and 0.053 µg of CMV-β-Gal as an internal control [49,50]. Transfection was carried out in low-serum RPMI 1640 medium (2% SFB and 2% HS) without antibiotics for 6 h, the medium was replaced, and the cells were treated with the indicated extracts and corresponding vehicle for 16 h. The medium was then discarded and the cells were lysed with reporter lysis buffer (Cell Culture Lysis buffer, Promega, Madison, WI, USA). The lysate was centrifuged, and the luciferase activity of the supernatant was determined using a luciferase assay system (Promega, Madison, WI, USA) and measured in a luminometer (Turner Biosystem 20/20). For normalization of luciferase activity, β-galactosidase activity was evaluated using o-nitrophenyl-β-D-galactopyranoside (ONPG) as the substrate in a standard colorimetric enzymatic assay. The luminescence signals obtained from the luciferase activity measurements were normalized to β-galactosidase colorimetric measurements to account for differences in the transfection efficiency or cell number. All transfection experiments were performed in triplicate.

#### 2.4.3. Adipocyte Differentiation

The 3T3-L1 cells were differentiated as described previously [51]. Briefly, 3T3-L1 cells were seeded in DMEM with 10% FCS at a density of 6 × 10^5^ cells/35 mm plate for lipid staining or 2 × 10^5^ cells/12-well plate for RNA extraction. After 48 h, the differentiation medium was used, containing freshly prepared solutions of 1 μg/mL insulin, 0.5 mM isobutylmethylxanthine (IBMX), and 0.1 µg/mL dexamethasone in the presence or absence of the test compound RGZ (1 µM), FMOC-Leu (25 µM), or Gracilex^®^ (60 µg/mL). After 48 h, the differentiation medium was replaced with DMEM containing 1 μg/mL insulin, then the medium was changed to DMEM supplemented with 10% FBS and P/S. Every 2 days the medium was changed. After 7 days of differentiation, the cells were harvested for RNA extraction using a HiBind column (Omega Bio-Tek, Norcross, GA, USA) following the manufacturer’s instructions. On the tenth day of differentiation, the cells were stained with Oil Red O, as described previously. Briefly, the cells were washed twice with PBS at 37 °C and then fixed with 4% paraformaldehyde (PFA) for 1 h at room temperature. The plate was washed with PBS and treated with 60% isopropyl alcohol for 6 min. After the plates were dried, Oil Red O (6:4) was added for 2 h at room temperature. Subsequently, the plates were washed extensively, dried, and photographed. For quantification of Oil Red O incorporation in cells, cells fixed in the plates were solubilized with pure isopropyl alcohol, and the absorbance of the solution was measured at 490 nm [52].

#### 2.4.4. MTT Viability Assay

A 3-(4,5-dimethylthiazol-2-yl)-2,5-diphenyl tetrazolium bromide (MTT) assay was performed in PC12 cells as described previously, with minor modifications [53]. Briefly, PC12 cells were seeded in 48-well plates at a density of 1.5 × 10^5^ cells/well in 250 µL of RPMI 1640 medium (10% HS, 5% SFB, and P/S) and grown for 24 h. Then, the medium was replaced with a low-serum RPMI 1640 medium (2% SFB, 2% HS, and Ab-Am).

The 3T3-L1 cells were seeded in 48-well plates at a density of 9 × 10^4^ cells/well in 250 µL of DMEM (10% CS and Ab-Am) and grown for 24 h. The cells were treated with the indicated extracts and corresponding vehicle for 16 h. The cells were washed with PBS and then incubated with 0.5 mg/mL MTT dissolved in a medium without phenol red or serum for 2 h at 37 °C. The MTT solution was discarded, and the cells were lysed with pure DMSO. The formazan product was measured in a spectrophotometer at 570 and 620 nm. The difference between the two measurements was used to calculate the percentage of cell survival with respect to the control.

#### 2.4.5. RT-qPCR

The RNA quality was evaluated by agarose gel integrity. Any contaminant genomic DNA was degraded by DNase treatment (RQ1, RNase-free DNase, Promega, Madison, WI, USA). The cDNA was generated using M-MLV reverse transcriptase (Promega, Madison, WI, USA) and random primers, while the qPCR was performed using SYBR Green (Brilliant II SYBR Green qPCR Mastermix, Agilent Technologies, Santa Clara, CA, USA) in a Stratagene MX-3000P qPCR System (Agilent Technologies). Fatty acid binding 4 (Fabp4) and lipoprotein lipase (Lpl) genes were amplified together with 3 reference genes in each run: glyceraldehyde-3-phosphate dehydrogenase (GAPDH), NoNo, and β-actin. The murine primer sequences used for Fabp4 were F: 5′-GCGTGGATTTCGATGAAATCA-3′ and R: 5′-CCCGCCATCTAGGGTTATGA-3′; for Lpl, F: 5′-ATTTGCCCTAAGGACCCCTG-3′ and R: 5′-GCACCCAACTCTCATACATTCC-3′; for GAPDH, F: 5′-TGCACCACCAACTGCTTAGC-3′ and R: 5′-GGATGCAGGGATGATGTTCT-3′; for NoNo, F: 5′-TGCTCCTGTGCCACCTGGTACTC-3′, R: 5′-CCGGAGCTGGACGGTTGAATGC-3′; and for β-actin, F: 5′-CCTGTGCTGCTCACCGAGGC-3′ and R: 5′-GACCCCGTCTCCGGAGTCCATC-3′.

### 2.5. Mouse Studies

#### 2.5.1. Mouse Treatments

C57BL/6J mice were maintained with water and food supplied ad libitum at the Pontificia Universidad Católica de Chile Catholic University animal facility. The mice were housed in a room with controlled temperature (24 °C) and humidity (55–60%) for a 12 h light and dark regime. All procedures were performed during the dark phase. Experimental protocols were approved by the biosafety and bioethics committee of the Pontificia Universidad Católica de Chile (project FONDEF#DO811031). All animal procedures were conducted according to the National Agency of Research and Development (ANID) guidelines.

To study Gracilex^®^ toxicity, 12 adult C57BL/6J mice (12 weeks old) from our animal facility were fed with a standard chow diet and corn oil (control) or Gracilex^®^ (300 mg/kg) for 15 days (see below in histopathological studies).

For high-fat diet experiments, 50 adult C57BL/6J mice (6 weeks old) were imported from the Jackson Laboratory (USA). At 6 weeks of age, the mice were separated into 5 groups of 10 animals: 4 groups were fed the high-fat diet (HFD: 60% kcal based on fat, Research Diets, Inc., New Brunswick, NJ, USA), and 1 group was fed the low-fat diet (LFD: 10% kcal based on fat) [54,55].

The animals were weighed twice a week throughout the study. At 12 weeks of age, the mice were treated as follows: control diet (LFD) mice were treated with corn oil, control HFD mice were treated with corn oil, positive control HFD mice were treated with rosiglitazone (5 mg/kg), and HFD mice were treated with two doses of Gracilex^®^, 90 and 300 mg/kg. The treatment was administered once a day by oral gavage for 30 days. The daily volume administered was approximately 50 µL.

#### 2.5.2. Measurement of Plasma Metabolic Parameters

Blood samples were obtained at the beginning and end of the treatment (12 and 16 weeks old, respectively). Before sampling, the mice were fasted for 15 h. Samples were obtained by puncture of the submandibular sinus, collected in heparinized tubes, and immediately centrifuged (3500 rpm/10 min at 4 °C) to obtain blood plasma.

Mouse adiponectin (DuoSet^®^ R&D Systems, MN, USA) and mouse insulin (Ultrasensitive EIA, ALPCO Diagnostic, NH, USA) were detected by ELISAs. Plasma levels of glucose and cholesterol were determined by enzymatic chemical kits (Wiener Laboratories SAIC, Buenos Aires, Argentina).

#### 2.5.3. Histopathological Studies

Twelve C57BL/6J mice (12 weeks old) per experimental group were randomly assigned to treatment with either corn oil (control) or Gracilex^®^ (300 mg/kg). The treatment was administered once a day by oral gavage for 15 days. The daily volume administered was approximately 50 µL. At the end of the treatment, the mice were euthanized by isoflurane overdose. The liver, kidney, esophagus, and stomach were extracted and maintained in 9.25% formalin (4 °C) for 48 h. For histopathological evaluation, samples were embedded in paraffin, processed under standard conditions, and stained with hematoxylin-eosin. Liver fat accumulation was determined by Sudan black staining, and liver glycogen deposition and kidney basal membrane were evaluated by Schiff’s periodic acid (SPA) staining. The histopathological findings were classified according to magnitude (0 = absent; I = slight; II = moderate; and III = severe).

### 2.6. C. Elegans Studies

Nematodes were cultivated at 20 °C under standard laboratory conditions on agar plates fed with *Escherichia coli* (OP50) as a food source [56]. The oxidative stress-sensitive *msra-1(tm1421)* mutant strain [57] was used for stress resistant assays. Prior to the oxidative stress challenge, the nematodes were fed for 24 h with Gracilex^®^ or its vehicle DMSO. For administration to nematodes, plates were prepared as follows: Gracilex^®^ was dissolved in DMSO at a concentration of 100 mg/mL. A stock of dead bacteria was prepared by growing the *E. coli* OP50 strain in lithium borate (LB) buffer overnight. After centrifugation, the pellet was resuspended to obtain a stock concentrated six times. This stock was heated at 70 °C for 30 min and stored at −20 °C until use. Then, 25 µg of Gracilex^®^ was mixed with 50 µL of dead bacteria and NP40 to a final concentration of 0.1% and a final volume of 52 µL. This mix was seeded on 2 mL of nematode growth medium (NGM) agar plates (35 mm). The plates were left to dry in the dark overnight in the laminar flow chamber and stored at 4 °C in the dark for no more than 1 week before use. For the oxidative stress test, L4 stage *msra-1* nematodes were grown for 24 h on plates with Gracilex^®^ or DMSO (vehicle). On the day of the test, the nematodes were transferred to a 24-well plate containing 600 µL of 9 mM hydrogen peroxide dissolved in M9 medium (minimal medium containing minimal salts: KH_2_PO_4_ 15 g/L, NaCl 2.5 g/L, Na_2_HPO_4_ 33.9 g/L, and NH_4_Cl 5 g/L). Ten nematodes were placed per well, and the movement was monitored every 10 min as described in [58] until there was no living nematode (approximately 2 h). Each experiment was performed in triplicate (30 nematodes per condition) [59].

### 2.7. Statistical Analysis

Cell-based assays for PPARγ transactivation were performed in triplicate and at least three independent experiments. The results were expressed as the mean fold change in induction relative to the control condition ±SEM. One-way analysis of variance (ANOVA) followed by Tukey’s multiple comparison test, was applied to determine statistical significance between the control and treatment conditions. The same statistical tests were used for the analysis of 3T3-L1 cell differentiation in the Oil Red O assay and in lipogenic gene mRNA induction as determined by qPCR. For the in vivo studies of insulin resistance, an unpaired *t*-test was used to compare the low-fat and high-fat diet groups. For analysis of the differences between subgroups of HFD-fed and LFD-fed mice, a one-way ANOVA and Tukey’s multiple comparison test were applied. Finally, in *C. elegans* in vivo experiments, a two-way ANOVA with repeated measures was used to compare differences in the survival curves of the nematodes fed Gracilex^®^ and those under control conditions. Fisher’s LSD post-test was used to determine significant differences at appropriate times. Significant differences were calculated at significance levels indicated as: * *p* < 0.5, ** *p* < 0.01, and *** *p* < 0.001. Statistical analysis was performed using GraphPad Prism 5.0 for Mac (GraphPad Software, San Diego, CA, USA). We analyzed the data for outliers using the Grubbs method (GraphPad Software). We detected just one outlier in the experimental point Gracilex^®^ + T007 of Figure 1C. Accordingly, we removed this point from the graph. We did not detect other outliers in the rest of the figures.

## 3. Results

### 3.1. Effect of Gracilex^®^ on PPARγ Transcriptional Activity

To determine whether Gracilex^®^ can induce transcriptional activation of PPARγ, we used two cell lines, HeLa and PC12, for transient transfection assays using the PPRE-tk-LUC reporter [60] cotransfected with the PPARγ expression vector (Figure 1A). In both cell types, Gracilex^®^ induced significant activity at 40 μg/mL, resulting in up to 65% of the maximal activation achieved by the full agonist rosiglitazone at 1 μM. To further confirm the activation of PPARγ by lipids derived from Gracilex^®^ and to study the contribution of PPARγ to the increased luciferase, we assessed the ability of Gracilex^®^ to activate the chimeric GAL4-DBD-PPAR-LBD protein from a Gal4-dependent MH100-Luc reporter using PC12 cells (Figure 1B). In this assay, the reporter was activated by the chimeric GAL4-DBD fusion protein, thus avoiding potential interference from any endogenous receptor [49,50]. We observed dose-response activation of PPARγ with Gracilex^®^. At a concentration of 100 μg/mL, Gracilex^®^ showed PPARγ activity induction comparable to that promoted by the SPARMs INT131 and FMOC-Leu at 25 μM (Figure 1B). In contrast, rosiglitazone-induced PPARγ activation was over 8 times higher at 1 μM compared to Gracilex^®^, which is consistent with the definition of a full agonist (Figure 1B). The PPARγ activation induced by Gracilex^®^ was prevented upon cotreatment with the PPARγ antagonist T0070907 [61], demonstrating that Gracilex^®^ specifically activates PPARγ (Figure 1C). These results suggest that Gracilex^®^ is a good source of natural ligands for PPARγ. No effects on the survival of PC12 cells were observed for any of the Gracilex^®^ concentrations evaluated (Appendix A).

SPPARMs are selective modulators of PPARγ activity with partial agonism and reduced side effects, such as adipose tissue gain [62]. Considering that PPARγ is a master gene for adipocyte differentiation and Gracilex^®^ activates PPARγ similarly to SPPARM at the higher doses evaluated (60 and 100 μg/mL), we next characterized the effect of Gracilex^®^ on the adipocyte differentiation of 3T3-L1 cells using a well-established protocol [51]. Incubation of the 3T3-L1 cell line with rosiglitazone (1 μM) and a hormonal mix for 7 days induced an evident increase in Oil Red O staining, which labels lipid droplets, indicating the accumulation of triglycerides [63]. Quantification of the isopropanol soluble fraction [52], derived from 3T3-L1 differentiated cells by absorbance at 490 nm, showed a twofold increase in cells treated with rosiglitazone compared to cells treated with only the hormonal mixture (Figure 2A,B). In contrast, neither FMOC-Leu or INT131 at 25 μM nor Gracilex^®^ at 60 μg/mL mimicked the effects of rosiglitazone (Figure 2A,B). To further confirm that SPPARMs and Gracilex^®^ would not induce adipose differentiation, we measured the induction of mRNA levels of the adipogenic marker genes lipoprotein lipase (LPL) and fatty acid binding protein 4 (FABP4). We observed that only rosiglitazone promoted the adipocyte differentiation program in 3T3-L1 cells (Figure 2C). To uncover the possible toxic effects of Gracilex^®^ in differentiated 3T3-L1 cells, we determined cell viability by MTT assay after Gracilex^®^, FMOC-Leu, INT131, or rosiglitazone treatment. No changes in cell survival were found for PPARγ agonists in 3T3-L1 cells (Appendix A). Taken together, these results indicate that Gracilex^®^ can induce PPARγ activation in a similar fashion to SPPARMs without promoting adipocyte differentiation (or toxic effects) in 3T3-L1 cells at the concentration used.

### 3.2. Effect of A. Chilense Oleoresin on Metabolic Dysfunction Caused by High-Fat Diet (HFD)-Induced Obesity in Male Mice

Although there are lean individuals with T2DM, obesity explains 60–80% of diabetes cases [38,39,40]. Even though animals do not develop T2DM as humans do, there are appropriate animal models for studying the effects of new natural products such as Gracilex^®^ on the development of T2DM induced by a high-fat diet. Studies indicate that a diet high in saturated fat (and not, for example, rich in sucrose) is the most appropriate for the development of metabolic dysfunction in murine models [64], imitating the physiological conditions associated with T2DM in humans, such as insulin resistance [45]. The C57BL/6J strain is a very useful strain for the development of diabetes induced by a high-fat diet (HFD) compared to other strains because it is genetically predisposed to develop obesity [54,65]. These animals are an ideal model for studying new therapeutic agents, including natural products, and their potential to modulate this condition [45]. Therefore, we used this model to test the potential use of Gracilex^®^ as a nutraceutical agent to improve HFD-induced insulin and glucose resistance. Initially, we compared body weight gain between a group of HFD-fed C57BL/6J mice and another group fed a normal diet (LFD). After 12 weeks, a significant difference (*p* < 0.001) in body weight was observed between the HFD and LFD groups, and was 27% greater in the high-calorie diet group than in the group receiving a normal diet (Figure 3A). When we compared biochemical blood parameters in LFD versus HFD mice, significantly increased glucose (81.7 vs. 119.5 mg/dL, respectively) (Figure 3B) and insulin (0.29 vs. 0.62 ng/mL, respectively) (Figure 3C) levels were observed, consistent with insulin metabolic dysfunction. Additionally, the plasma adiponectin levels were markedly decreased in the HFD mice compared to the LFD mice (3.8 vs. 1.8 μg/mL, respectively) (Figure 3D), confirming the parameters described in obese animals with insulin resistance. Finally, as a result of high fat intake, the plasma cholesterol values increased significantly in the HFD compared to LFD mice (121 vs. 152 mg/dL, respectively) (Figure 3E). Taken together, these results indicate the effective induction of the insulin resistance state by HFD intake in mice, which is the condition required to test the effect of Gracilex^®^.

To evaluate whether possible harmful effects of Gracilex^®^ ingestion would be observed in mice, we performed a histopathological study in mice treated for a shorter period of time (15 days). No major changes in the kidney and stomach were found (data not shown). In the liver of mice fed a normal chow diet, we observed moderate glycogen accumulation by PAS staining and no changes in fat accumulation (Appendix A). Additionally, no changes in body weight were found (Appendix A). These results suggest no toxic effects were induced by Gracilex^®^.

The HFD-treated mice were then subdivided into four groups, maintained on the HFD and orally treated for 30 days with rosiglitazone at 5 mg/kg per day and two Gracilex^®^ doses, 90 and 300 mg/kg per day, or corn oil as a vehicle. After the treatment period, no variations in body weight were observed between subgroups; however, differences between the HFD and LFD groups were maintained (Figure 4A), as demonstrated previously [54]. The analysis of blood parameters at the end of the treatment period showed that the mice receiving the HFD and corn oil maintained a significant increase in glycemia compared with the controls (Figure 4B). In the HFD-fed mice, the group treated with Gracilex^®^ 300 mg/kg showed a complete recovery of blood glucose levels, reaching the levels of LFD control mice (Figure 4B). Rosiglitazone and Gracilex^®^ (90 and 300 mg/kg) were also able to decrease glucose to control levels (Figure 4B). In relation to insulin parameters, the HFD mice treated with Gracilex^®^ at a dose of 300 mg/kg showed a significant decrease in insulin, reaching values close to 50% of the concentrations found in the insulin-resistant HFD-treated control group (Figure 4C). However, as expected [55,66], synthetic rosiglitazone promoted a decrease in insulin levels, normalizing this parameter to that observed in the control LFD mice.

Adiponectin levels completely recovered after 30 days of rosiglitazone treatment in the HFD-control mice compared with the HFD-rosiglitazone mice, which was equivalent to the values measured in the LFD control group (Figure 4D). Unexpectedly, Gracilex^®^ at 300 mg/kg per day had the opposite effect to rosiglitazone, with a significant reduction in adiponectin levels compared the HFD-control mice and HFD-Gracilex^®^ (300 mg/kg) mice (Figure 4D), which is consistent with the fact that Gracilex^®^ does not induce adipocyte differentiation in 3T3-L1 cells as rosiglitazone does (Figure 2). Finally, plasma cholesterol was not affected by treatment with either rosiglitazone or Gracilex^®^ in the HFD-treated mice (Figure 4E), and this group maintained high levels compared to the mice fed a normal diet. These data demonstrate that Gracilex^®^ improves insulin sensitivity and normalizes glucose levels induced by HFD with no alterations in weight gain, despite the reduced levels of circulating adiponectin, which is an adipose tissue-derived hormone that regulates the expansion and healthy function of adipose tissue [67]. According to our studies, it is possible to consider Gracilex^®^ as an alternative source of natural PPARγ ligands with similar properties found in selective PPAR modulators, presenting partial receptor activation that is sufficient for insulin sensitization in HFD-induced metabolic dysfunction with no effect on the promotion of adipocyte differentiation.

Species of the *Gracilaria* genus have already been studied in terms of lipid components by several authors, highlighting the rich and unique composition of sulfoglycolipids, polyunsaturated fatty acids, oxidized fatty acids, pigments, and lipophilic antioxidant molecules [13,15,68,69,70]. An overall analysis of the fatty acid content of Gracilex^®^ is shown in Table 1 as the relative abundance (%) of total fatty acid methylesters detected by gas chromatography with a flame ionization detector (GC-FID). We found a high content of saturated fatty acids (51.4 ± 4.89%), followed by polyunsaturated (25.7 ± 3.12%) and monounsaturated (19.6 ± 2.88%) fatty acids. The most abundant fatty acids were palmitic acid (C16:0), which contributed 40% of the total fatty acids analyzed; arachidonic acid (C20:4, n-6) with 21.1%; oleic acid (C18:1, n-9) with 14.13%; myristic acid (C15:0) with 4.4%; and 11-octadecenoic acid (C18:1, n-7) with 4.0% (Table 1). In general, we found that this distribution is similar to the fatty acid profile reported by Da Costa et al. [13] in their methanolic extract of *Gracilaria* sp., a red seaweed closely related to *A. chilensis*. The omega-3 fatty acid content was low, reaching up to 1.22% of total fatty acids analyzed. In contrast, the total content of omega-6 fatty acids was higher (Table 1). Arachidonic acid is the precursor of eicosanoids and oxylipins, bioactive lipidic promoters of the defense response in seaweed [12,14]. Most natural and endogenous PPARγ ligands are derived from arachidonic acid metabolization [29]; therefore, the increased abundance in Gracilex^®^ may account for its high PPARγ transactivation capacity.

The nutritional value of red macroalgae is also due to the diverse content of pigments and antioxidants [69,71,72]. Consequently, we determined the total antioxidant capacity (TAC) using CUPRAC assay that was compatible with lipophilic antioxidants (such as tocopherols) [47] in Gracilex^®^ and two botanical lipid extracts, spirulin and maqui, produced with dichloromethane extraction as Gracilex^®^. Spirulin and maqui berries are known functional foods with high nutraceutical value. Spirulin is a cyanobacterium with known antioxidant and nutritional beneficial effects [73], and maqui berry (*Aristotelia chilensis*), a flavonoid-rich berry, is well known for its antioxidant and insulin-sensitizing effects [74]. Table 2 shows that Gracilex^®^ is slightly superior to the other botanical oleoresins, reaching a mean of 430 mg uric acid equivalent per 100 mg of oleoresin. To further explore the antioxidant molecules, we evaluated the content of tocopherols and β-carotene in Gracilex^®^. As indicated in Table 3, we found high levels of β-carotene and tocopherols, but did not detect lycopene. The β-carotene content was approximately 1538 μg/g Gracilex^®^, and the total tocopherol content was 6673 μg/g Gracilex^®^, which was distributed as follows in the different types of tocopherols: α-tocopherol (527.7 μg/g), γ-tocopherol (5332.8 μg/g), and ∂-tocopherol (2660 μg/g). Thus, the antioxidant capacity of Gracilex^®^ is partly due to the high levels of β-carotene and tocopherols, particularly γ-tocopherol.

### 3.3. Antioxidant Properties of Gracilex^®^

Considering the high tocopherol and β-carotene content, we next assessed the ability of Gracilex^®^ to confer antioxidant resistance in vivo. The oxidative stress-sensitive *msra-1* mutant strain of *C. elegans* [57] was chosen as a model to evaluate in vivo oxidative stress resistance. Since this strain has been proven to be very sensitive to acute oxidative treatment, it is a powerful tool to assess the physiological consequences of oxidative stress in vivo [75,76]. Taking advantage of the autofluorescence of pigments contained in Gracilex^®^, we first evaluated whether the nematodes were able to ingest it in a mixture with dead OP50 bacteria. As shown in Figure 5A, Gracilex^®^ was visualized along the digestive tracts of the nematodes (highlighted by red arrows), indicating that they can ingest it. For antioxidant resistance assays, *msra-1* L4 larval stage nematodes were fed for 24 h with DMSO (control), Gracilex^®^, or N-acetylcysteine (NAC) mixed with dead bacteria. Then, the worms were challenged with a hydrogen peroxide solution and their movement was scored every 10 min. Figure 5B shows the percentage of active nematodes over time after the challenge. After 80 min, 60% of the *msra-1* mutants were already inactive compared to 40% of the nematodes fed Gracilex^®^. From 70–110 min, we consistently found more active nematodes, as shown by the right shift in the survival slope, which was significant at 80 min. Similar behavior was observed in the nematodes previously exposed to 5 mM NAC, a well-known potent antioxidant, which was demonstrated to rescue the unbalance in the redox homeostasis induced by hydrogen peroxide in the *msra-1* mutant [57]. The time response curve in Figure 5B showed the highest protection of Gracilex^®^ at 80 min. This time response is in the range of NAC, between 70 to 90 min after the peroxide challenge (Figure 5C,D). These findings indicated that nematodes fed with Gracilex^®^ had increased resistance against oxidative stress, improving their survival, in agreement with the CUPRAC analysis and the characterization of their antioxidant components.

## 4. Discussion

Interest in the study of natural bioactive molecules able to activate PPARγ has grown over the years due to the importance of the physiological processes that these receptors regulate and the side effects caused by pharmacological PPARγ drugs. This interest is parallel to an increase in the study of the biomedical potential of seaweed due to the large amount of bioactive molecules that they possess [9,78,79,80]. The study of the lipid components of *A. chilense* and other closely related red seaweeds demonstrated the presence of PUFAs and eicosanoids [14,16,27]; however, whether the lipids present in the *Agarophyton* genus were able to activate PPARγ and had insulin sensitizer and antioxidant effects in vivo was unknown. Here, we demonstrated that Gracilex^®^ includes ligands that are able to activate PPARγ at a similar level to that observed with the SPPARMs FMOC-Leu and INT131. In addition, we found a beneficial effect of Gracilex^®^ on insulin sensitization in a mouse model of HFD-induced metabolic disorder with a lack of some of the side effects described for the full agonist rosiglitazone on adipose differentiation.

PPARγ is a lipid sensor of the nutritional and metabolic status of an organism [78]. As a lipid sensor, this receptor is activated by structurally diverse lipophilic substances because the PPARγ ligand-binding pocket is the largest among the nuclear receptors, allowing entry and binding of up to two different molecules in a covalent or noncovalent manner [81,82,83,84]. Our first goal was to evaluate whether the lipids present in Gracilex^®^ were able to activate PPARγ by comparing it with the full agonist rosiglitazone and two synthetic SPPARMs, FMOC-Leu and INT131 [62,85]. The magnitude of the transcriptional induction of PPARγ achieved by Gracilex^®^ was similar to that observed for SPPARM agonists, revealing, as the first finding, that Gracilex^®^ possesses SPPARM-like ligands. The fact that similar induction of PPARγ transcriptional activity was induced in two different cell lines (Figure 1) with a dose-dependent response suggests that the main bioactive molecules are already present in the extract and that the intrinsic cell metabolism may not undergo further modifications.

Natural extracts presenting PPARγ agonist activity were reported and reviewed extensively by Wang et al. [37]. In a broad screening of extracts derived from 71 plant species, almost 40% (28 extracts) presented PPARα or PPARγ activity. However, whether this represents insulin sensitizer activity or antioxidant activity in vivo was not evaluated. In agreement with our findings, greater biological activity was found in extracts produced with dichloromethane than in those obtained with methanol [80], suggesting that the putative PPAR ligands are more easily extracted with apolar solvents. Similar screening was performed by Kim et al. [86] using different macroalgae and methanol/ethyl acetate extractions. They found that the extract from *Sargassum yezoense*, a brown seaweed, showed the strongest potency for PPARα/γ transcriptional response in a PPRE-based transcriptional assay. According to their studies, the increased PPAR activity was explained by the presence of sarquinoic and sargahydroquinoic acids that could increase adipocyte differentiation of 3T3-L1 cells, which is an unwanted side effect of natural derived ligands expected to be used in diabetes, since this feature is associated with weight gain in T2DM patients [87]. Gracilex^®^ not only induced partial PPARγ activation similar to SPPARMs, but also improved insulin sensitivity and normalized glucose levels with no alterations in weight gain, and did not promote adipocyte differentiation in a classical in vitro model assay.

A recent analysis of *A. chilense* lipids by Honda et al. [14] revealed that glycerolipids comprise the largest proportion of the total lipids (monogalactosyldiacylglycerol and phosphatidylcholine), carrying mostly a combination of arachidonic and palmitic acids (20:4n-6/16:0) in their structure. In agreement with this study, our analysis of the fatty acid profile of Gracilex^®^ showed that palmitic and arachidonic acids were the most abundant fatty acids, representing 40 and 21.1%, respectively. Our findings also agree with a recent report on the fatty acid profile of the related red macroalga *Gracilaria* spp. [13]. The involvement of these fatty acids and their metabolites in the ability of Gracilex^®^ to induce PPARγ activation is currently unknown and under investigation in our laboratory. Therefore, the biological activities that we describe in this work cannot be directly attributed to an increase in the transcriptional activity of PPARγ.

One of the central pathologies in metabolic syndrome is insulin resistance, an event that occurs before T2DM. The development of this pathology directly correlates with overweight and obesity; indeed, obesity explains 60–80% of diabetes cases [88,89]. Here, we provide evidence that Gracilex^®^ treatment might be of benefit for overweight patients with insulin resistance, although its effectiveness in humans requires further investigation. Oral treatment of HFD-fed mice with Gracilex^®^ for one month showed a notable decrease in glucose and insulin levels in the blood altered by the HFD at the 300 mg/kg dose, similar to rosiglitazone at 5 mg/kg. Rosiglitazone increased adiponectin levels, as expected [90]. On the contrary, Gracilex^®^ significantly decreased adiponectin levels in HFD-fed mice compared to HFD-fed control mice.

These findings have several interpretations, first, it is possible that natural lipids directly activate PPARγ similar to SPPARMs; however, this possibility was not evaluated in our study and deserves further investigation. Moreover, the anti-oxidant effect of Gracilex^®^ (see below) may contribute to the improvement observed in metabolic dysfunction since it is known that the generation of reactive oxygen species contributes to the progression of T2DM [91].

On the other hand, this finding indicates that the beneficial effect of Gracilex^®^ is not due to the reported capacity of PPARγ agonists to increase adiponectin levels [90]. Adiponectin is a hormone secreted by adipose tissue and has autocrine and endocrine actions with reported global insulin sensitizer effects [67]. Nonetheless, adiponectin increases the growth of fat tissue, and the overexpression of adiponectin in 3T3-L1 cells increases lipid storage and adipogenesis [92]. Thus, the limited capacity of Gracilex^®^ to increase adipocyte differentiation and accumulation of triglycerides measured by Oil Red O staining in 3T3-L1 cells correlates with the decreased adiponectin levels in vivo induced by Gracilex^®^ in HFD-fed mice. Further experiments studying the genetic programs and transcriptomes in 3T3-L1 cells and adipose tissue from HFD-fed mice treated with the vehicle and Gracilex^®^ will help elucidate this issue [93].

As a source of multiple types of chemical molecules, Gracilex^®^ showed antioxidant activity, as was previously reported in diverse algae-derived extracts. While aqueous extracts have been characterized in depth for either their in vivo activity or their antioxidant components (such as phenols and polyphenols, amino acid-like mycosporines, and sulfated polysaccharides), scarce information exists for the apolar organic-produced fraction, like the one we produced in this study. In the red seaweed *Gracilaria blodgetti,* a chloroform extract could decrease the oxidative stress induced in leukocytes [94,95,96,97]. These results indicate that apolar extracts can be a source of antioxidants for biological membranes.

Depending on the type of alga, solvent used, and season of harvest, a majority or minority of antioxidant components and their activities can be redistributed between polar or apolar fractions [11,98]. Difficulties in determining the in vitro capacity of apolar extracts or molecules through classical methods are well known, and more than one assay should be used to confirm their antioxidant potential [99,100]. Considering the few alternative methods for determining antioxidant capacity in oleoresins, in addition to the in vitro CUPRAC method, we evaluated the in vivo antioxidant capacity of Gracilex^®^ using a mutant strain of *C. elegans* that is sensitive to oxidative stress [57]. We found that nematodes fed Gracilex^®^ were protected against exposure to hydrogen peroxide at a similar magnitude as that given by N-acetylcysteine treatment, reflected by an increase in nematode survival. N-acetylcysteine is a source of glutathione production, and has been shown to prevent reactive oxygen species generation [77] and increase oxidative stress protection in *C. elegans* [76]. Therefore, it was an appropriate positive control for our experiments. These results correlate with high content of tocopherols and β-carotene, well-known antioxidants that reduce lipid peroxidation, in Gracilex^®^ [101,102,103].

## 5. Conclusions

Altogether, our results indicate that Gracilex^®^, an oleoresin produced using organic solvent extraction from the red macroalgae *A. chilense,* contains natural PPARγ ligands that increase PPARγ transcriptional activity. Like SPPARMs, Gracilex^®^ did not induce preadipocyte differentiation like the PPARγ receptor full agonist rosiglitazone. Furthermore, HFD mice fed Gracilex^®^ showed normalization of glucose and insulin parameters altered by the diet without increased adiponectin. This result was consistent with the idea that the natural lipids present in Gracilex^®^ act as SPPARM-like agonists of PPARγ. However, direct evidence that Gracilex^®^ exerts its biological effects through PPARγ activation warrants future research. Moreover, Gracilex^®^ showed high concentrations of tocopherols and β-carotene, which correlated with the antioxidant effect in vivo using a *C. elegans* model of oxidative stress. Therefore, antioxidant activities may also contribute to the normalization of insulin and glucose parameters altered by the HFD. Altogether, our results indicate that Gracilex^®^ represents a good source of natural PPARγ ligands and antioxidants with the potential to mitigate metabolic disorders. Nevertheless, its validation in humans is required to ensure its high nutritional value.

## Figures and Tables

**Figure 1 nutrients-13-01828-f001:**
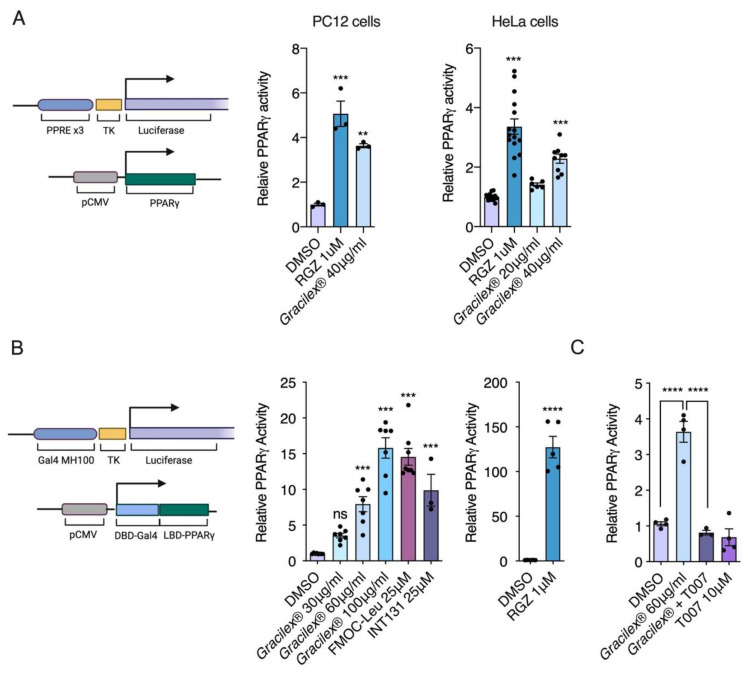
***Agarophyton chilense* oleoresin** (Gracilex^®^) induces PPARγ transcriptional activation. Two cell-based reporter assays to measure PPARγ activation were performed. Left panel (**A**): schematic representation of vectors used for cotransfection. PC12 and HeLa cells were cotransfected with plasmids expressing full-length murine PPARγ and a reporter plasmid containing PPRE (peroxisomal proliferator response element) driving expression of luciferase, together with plasmid expressing beta-galactosidase as transfection control. Full PPARγ agonist rosiglitazone (RGZ, 1 µM) and Gracilex^®^ (40 µg/mL) induced transcriptional activation of PPARγ, as shown by increased luciferase activity normalized to beta-galactosidase expression (expressed as relative PPARγ activity). In PC12 cells (*n* = 3) and HeLa cells (*n* = 6–13), treatments (RGZ and Gracilex^®^) were compared to control transfected cells (DMSO), and values are presented as ±SEM, ** *p* < 0.01, and *** *p* < 0.001 (one-way ANOVA and Tukey’s post hoc test). Left panel (**B**): schematic representation of vectors used for cotransfection. Plasmid with expression of a chimeric construct of PPARγ ligand binding domain (LBD-PPARγ) gene and DNA binding domain of Gal4 (DBD-Gal4) was cotransfected with a plasmid possessing the response element of GAL4 (Gal4 MH100) driving luciferase expression. As in (A), a third plasmid was included with expression of beta-galactosidase as transfection control. Increased PPARγ activation is expressed as relative PPARγ activity. Gracilex^®^ induced dose-response activation of PPARγ is comparable to activation induced by selective PPAR modulators (SPPARMs) of PPARγ, FMOC-Leu, and INT131 at a concentration of 25 µM. Instead, RGZ at 1 µM shows much higher activation (110 times over control), as expected for a full agonist. Values are presented as ±SEM, *n* = 3–7, and *** *p* < 0.001, **** *p* < 0.0001 left panel: one-way ANOVA and Tukey’s post hoc test; right panel: unpaired *t*-test). The (**C**) PC12 cells were transfected as described in (**B**) and treated with Gracilex^®^ (60 µg/mL) with or without PPARγ antagonist T0070907 (T007, 10 µM). Values are presented as ±SEM, *n* = 4, and **** *p* < 0.0001 (one-way ANOVA and Tukey’s post hoc test).

**Figure 2 nutrients-13-01828-f002:**
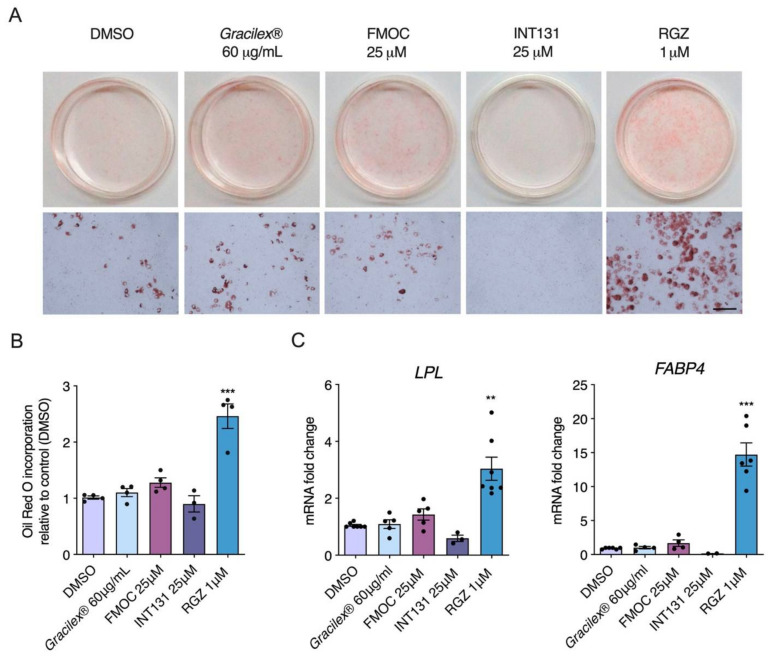
Adipogenesis in 3T3-L1 cells induced by rosiglitazone but not Gracilex^®^ or PPARγ SPPARMs, FMOC-Leu (FMOC), and INT131. First, (**A**) 3T3-L1 preadipocytes were cultured with or without PPARγ agonist FMOC-Leu and INT131 (25 μM), rosiglitazone (RGZ, 1 μM), and Gracilex^®^ (60 μg/mL) as specified in the methodology section. After 10 days of culture initiation, cells were stained with Oil Red O to label triglycerides, and the plates were photographed. Upper panels: microphotography of 35 mm plates. Lower panels: higher magnification of differentiated adipocytes stained with Oil Red O. Scale bar, 50 μm. The (**B**) quantification of Oil Red O incorporation was measured after cell lysis with isopropanol at a wavelength of 490 nm. Values are expressed as mean relative to that of control cells (treated with DMSO) ±SEM, *n* = 6, *** *p* < 0.001 (one-way ANOVA and Tukey’s post hoc test). The (**C**) effects of Gracilex^®^ and PPARγ agonists on expression of lipogenic pathway-related genes during differentiation process of mouse 3T3-L1 preadipocytes were measured after 7 days of culture initiation, as indicated in the methodology section. After treatment with each compound, cells were lysed and mRNA levels of LPL and FABP4 genes were measured by qPCR. The graph shows the relative abundance compared to the control (untreated cells, control DMSO). Values are expressed as the mean relative to that of control cells (treated with DMSO) ±SEM, *n* = 4, ** *p* < 0.01, *** *p* < 0.001 (one-way ANOVA and Tukey’s post hoc test).

**Figure 3 nutrients-13-01828-f003:**
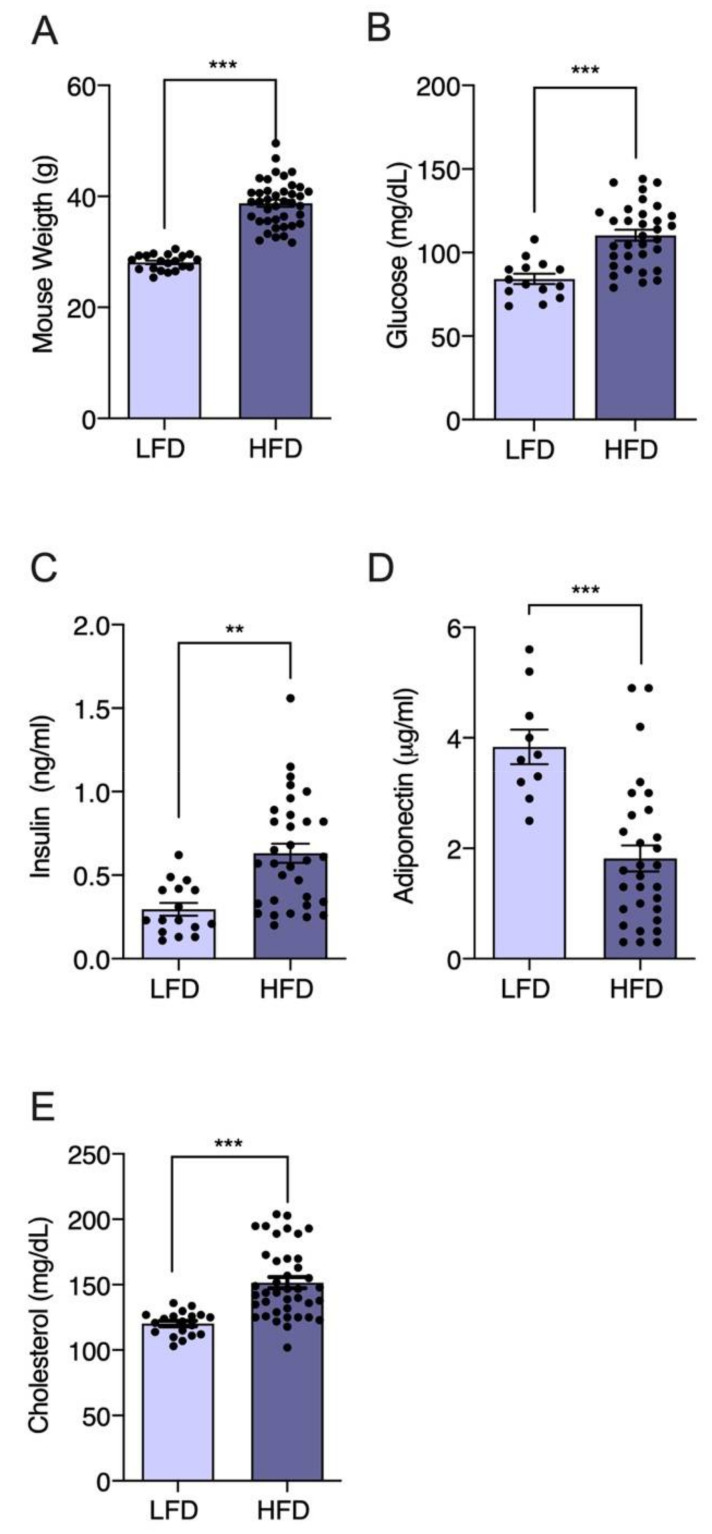
Promotion of weight gain and glucose and insulin resistance in male C57BL/6J mice administered a high-fat diet (HFD). Six-week-old mice were treated for an additional 6 weeks with a low-fat diet (LFD, *n* = 20) and HFD (*n* = 40). The (**A**) weight recording in groups of mice given LFD and HFD after 6 weeks of treatment. During the whole period, weights were recorded twice a week. Values are expressed as mean recorded weight ±SEM, *** *p* < 0.001 (unpaired *t*-test). The (**B**–**E**) biochemical blood parameters were measured in both groups: plasma glucose (mg/dL), plasma insulin (ng/mL), plasma adiponectin (μg/mL), and plasma cholesterol (mg/dL). Values are expressed as mean ± SEM, and significant differences are highlighted between LFD and HFD groups. ** *p* < 0.01 and *** *p* < 0.001 (unpaired *t*-test).

**Figure 4 nutrients-13-01828-f004:**
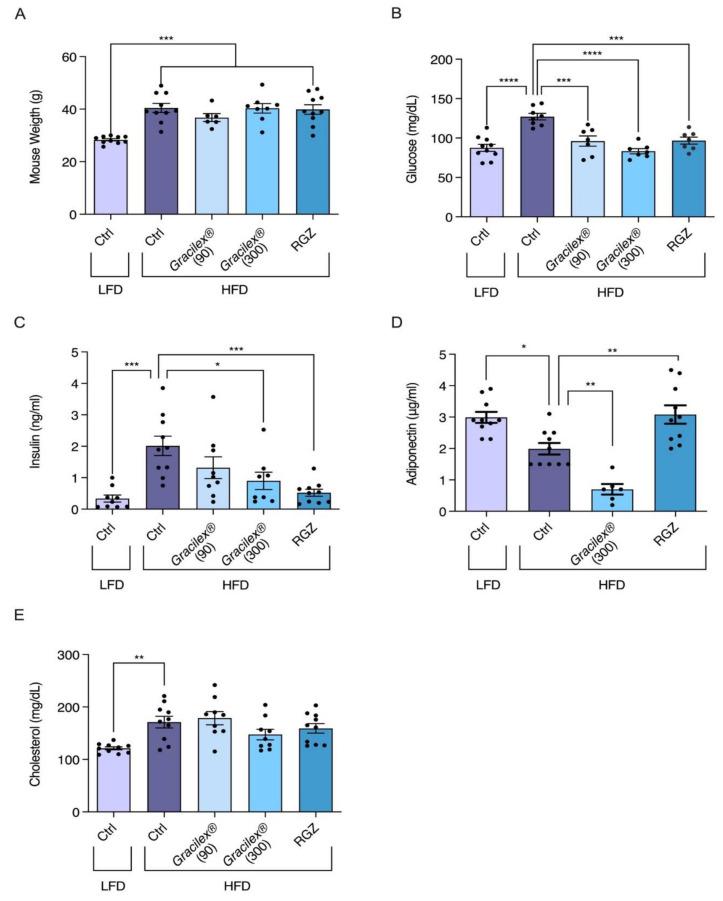
Gracilex^®^ improves altered glucose and insulin parameters in the plasma of male C57BL/6J mice given a HFD. After 6 weeks of LFD and HFD treatment, as indicated in Figure 3, LFD mice continued with the same diet and additional daily treatment with approximately 50 µL of corn oil (vehicle for drugs) for 1 month (30 days). HFD-treated mice were divided into 4 groups and continued with the same diet, but 4 daily treatments were given: corn oil (vehicle), 90 and 300 mg/kg of Gracilex^®^, and rosiglitazone (RGZ, 5 mg/kg). During the whole period, weights were recorded twice a week. The (**A**) comparison of weights at the beginning and end of treatment for each group. Values are expressed as mean ± SEM, *n* = 10, *** *p* < 0.001 (one-way ANOVA and Tukey’s post hoc test). (**B**–**E**) After treatment, mice were fasted for 15 h before taking blood samples; plasma glucose (mg/dL), plasma insulin (ng/mL), plasma adiponectin (μg/mL), and plasma cholesterol (mg/dL) values are shown. Values are expressed as mean ± SEM, *n* = 10. Significant differences are highlighted between LFD and HFD groups (left panels: ** *p* < 0.01, *** *p* < 0.001, unpaired *t*-test). Significant differences found in post-treatment experiments in HFD group were related to their HFD control. * *p* ≤ 0.05, ** *p* < 0.01, and *** *p* < 0.001 (one-way ANOVA and Tukey’s post hoc test).

**Figure 5 nutrients-13-01828-f005:**
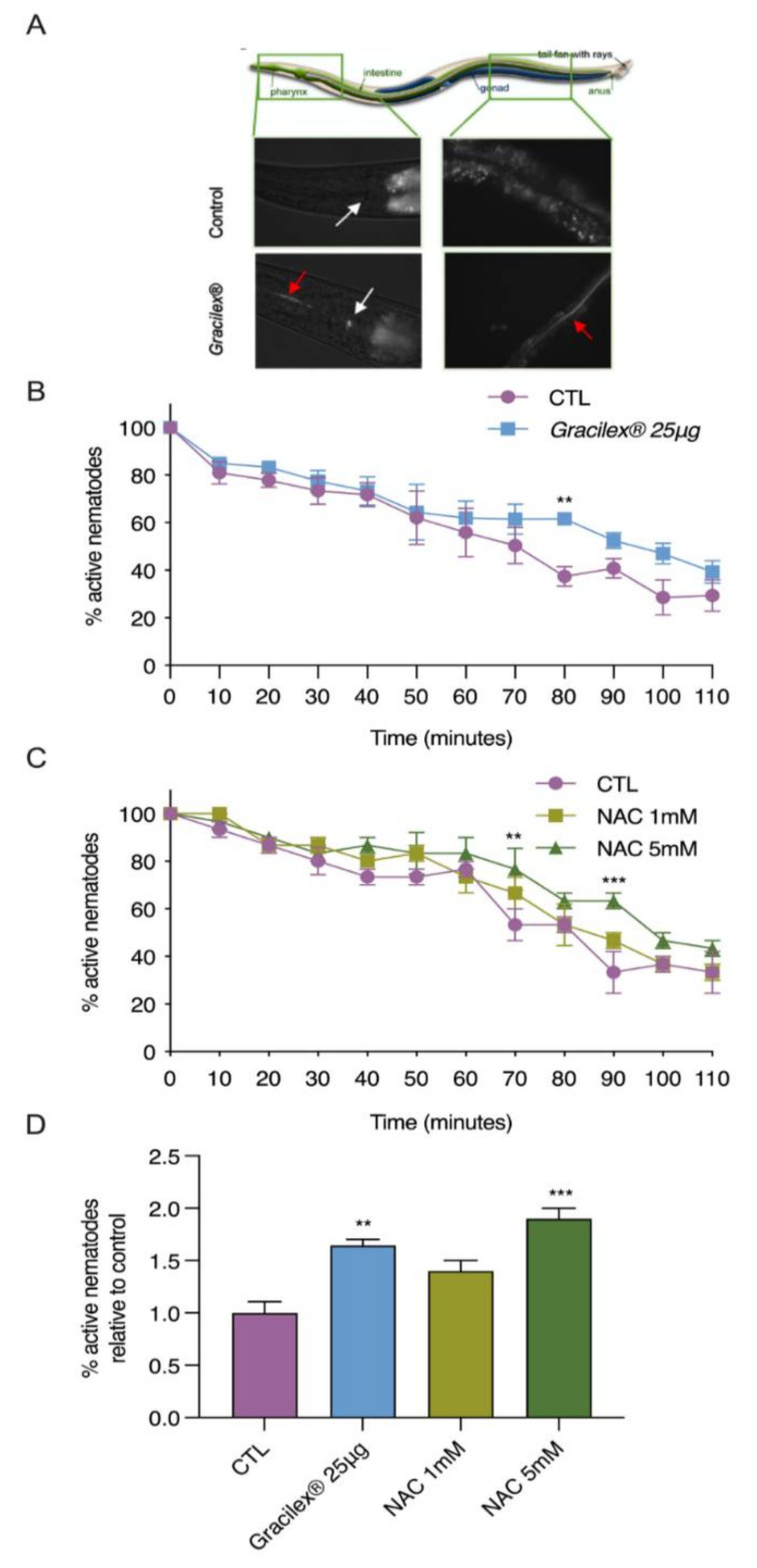
Gracilex^®^ increases oxidative stress resistance in *C. elegans* challenged with hydrogen peroxide. A (**A**) representative illustration of whole *C. elegans* organism from Corsi et al., 2015 [77]. Red autofluorescence of extract was observed along nematode digestive tract visualized under fluorescence microscopy (magnification 400×). Gracilex^®^ was dissolved in DMSO and 25 μg was mixed with 25 µL of dead bacteria and dispersed on 1.5 mL of agar plate (35 mm). White arrow shows nematode pharynx; red arrow highlights fluorescent extract inside nematode. Control (CTL) corresponds to adult N2 worm grown under standard conditions. The (**B**) L4 stage nematodes from the *msra-1* strain were fed Gracilex^®^ for 24 h then exposed to oxidative stress by 9 mM hydrogen peroxide for up to 110 min. Values are expressed as the mean (*n* = 5) percentage of survival ±SEM, ** *p* < 0.01 (two-way ANOVA with repeated measures and Fisher’s LSD test). The (**C**) same experimental assay was performed as described in (B), but nematodes were fed two concentrations (1 and 5 mM) of N-acetylcysteine (NAC) dissolved in water. NAC was used as the positive control. Values are expressed as mean (ρρρ 3) percentage of survival ±SEM, ** *p* < 0.01 and *** *p* < 0.001 (two-way ANOVA with repeated measures and Fisher’s LSD test). Significant differences in the percentage of active nematodes were found under treatment with 5 mM NAC compared to the control at 70, 90, and 120 min after exposure to hydrogen peroxide. The (**D**) time points where maximum protection effect was observed in B and C were plotted relative to each control ** *p* < 0.01 and *** *p* < 0.001 (one-way ANOVA and Tukey’s post hoc test) to facilitate comparison.

**Table 1 nutrients-13-01828-t001:** Fatty acid composition (relative abundance, % over total fatty acids) of *Agarophyton chilense* oleoresin (Gracilex^®^). Fatty acids were detected as methyl fatty acids by GC-FID. Results are expressed as mean and standard deviation (SD) of the mean of six independent Gracilex^®^ preparations. Fatty acids were detected as methyl fatty acids by GC-FID. Mean of six lipid extracts produced from independent algal biomass harvested and cultured in different seasons (fall, winter, and summer).

	Fatty Acid	Chain Length	Mean %	SD
Saturated	Decanoic Acid	10:00	0.760	1.1
	Dodecanoic Acid	12:00	0.305	0.3
	Tridecanoic Acid	13:00	0.983	0.3
	Tetradecanoic Acid	14:00	4.438	0.9
	Pentadecanoic Acid	15:00	0.440	0.3
	Hexadecanoic Acid	16:00	40.005	5.4
	Heptadecanoic Acid	17:00	0.750	1.3
	Octadecanoic Acid	18:00	2.683	3.0
	Eicosanoic Acid	20:00	0.152	0.08
	Docosanoic Acid	22:00	0.238	0.05
	Tetracosanoic Acid	24:00	0.127	0.08
Mono-Insaturated	10-Pentadecaenoic Acid	15:1 n-5	1.87	1.8
	9-Hexadecaenoic Acid	16:1 n-7	0.46	0.3
	9-Octadecaenoic Acid	18:1 n-9	14.13	4.6
	11-Octadecaenoic Acid	18:1 n-7	4.07	1.4
Omega-6 Polyunsaturated	9,12-Octadecadienoic Acid	18:2 n-6	2.87	0.81
	6,9,12-Octadecatrienoic Acid	18:3 n-6	0.17	0.10
	11,14-Eicosadienoic Acid	20:2 n-6	0.34	0.37
	8,11,14-Eicosatrienoic Acid	20:3 n-6	0.47	0.09
	5,8,11,14-Eicosatetraenoic Acid	20:4 n-6	21.06	3.81
Omega-3 Polyunsaturated	9,12,15-Octadecatrienoic Acid	18:3 n-3	0.390	0.42
	5,8,11,14,17-Eicosapentaenoic Acid	20:5 n-3	0.408	0.38
	7,10,13,16,19-Docosapentaenoic Acid	22:5 n-3	0.27	0.12
	4,7,10,13,16,19-Docosahexaenoic Acid	22:6 n-3	0.125	0.04
Other	Conjugated Fatty Acids			
	Fatty Acid	Chain Length	Mean %	SD
	c9, t11-octadecadienoic Acid	18:2 n-cla	0.187	0.06
	Trans Fatty Acids			
	Fatty Acid	Chain Length	Mean %	SD
	10-Transpentadecaenoic Acid	15:1 n-5t	0.258	0.24
	9-Octadecaenoic Acid	18:1 n-9t	1.037	1.30
	11-TransOctadecaenoic Acid	18:1 n-7t	0.540	0.30
	9,12-Octadecadienoic Acid	18:2 n-6tt	0.11	0.04

**Table 2 nutrients-13-01828-t002:** Total antioxidant capacity of oily extracts derived from botanical sources measured with CUPRAC assay. Results were standardized using a uric acid curve according to the manufacturer’s instructions and expressed as equivalent of uric acid per 100 mg of oleoresin. Results are expressed as the mean of independent determinations ±SEM. Gracilex^®^ (*n* = 7), spirulina oleoresin (*n* = 3), and maqui oleoresin (*n* = 3).

Sample	mg Uric Acid Eq/100mg Oleoresin
	Mean ± SEM
*Gracilex* ^®^	430 ± 58.3
Spirulina oleoresin	344 ± 90.6
Maqui oleoresin	305 ± 49.9

**Table 3 nutrients-13-01828-t003:** Characterization of tocopherol and β-carotene content of Gracilex^®^. Results are expressed as mean of six independent extract preparations ±SEM.

Sample	μg/g of Gracilex^®^
	Mean ± SEM
α-Tocopherol	527.7 ± 85.3
γ-Tocopherol	5332.8 ± 1523.3
δ-Tocopherol	2660 ± 397.1
Total Tocopherols	6673 ± 1568.2
β-Carotene	1538 ± 378.4

## Data Availability

Not applicable.

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
