# Peer review of "Characterization of an Agarophyton chilense Oleoresin Containing PPARγ Natural Ligands with Insulin-Sensitizing Effects in a C57Bl/6J Mouse Model of Diet-Induced Obesity and Antioxidant Activity in Caenorhabditis elegans"

_nutrients, 2021, doi:10.3390/nu13061828_

Round 1

Reviewer 1 Report

Comments ot paper Nutruents Characterization of an agarophyton chilense oleoresin containing pparγ natural ligands with insulin-sensitizing effects in a c57bl/6j mouse model of diet-induced obesity and antioxidant activity in caenorhabditis elegans.

Excellent work

Only small things to explain some topics a little better.

In the Title: Agarophyton, capital letter as Caenorhabditis as well

«in» is not italic

It would be interesting to write in a clear way the objectives of the work at the end of the Introduction section.

It is not clear what «Gracilex®» is. It is a patent to produce an oleoresin? A certain mixture of lipids with established chemical composition? Has this oleoresin a chemical defined structure? In line 200 says that fatty acids were determined, so the Gracilex has not a chemical defined structure, is it? 

Line 199: the name in italic

Lien 208: it should be advisable to describe the number of the ATTC cell culture lines.

Conclusions can be improved, more conclusions can be withdrawn from so many tests, for sure.

Author Response

Answer to Reviewer 1.

Point 1: In the Title: Agarophyton, capital letter as Caenorhabditis as well «in» is not italic. We have capitalized PPARγ and Caenorhabditis and “in” is not in italic.

Point 2: It would be interesting to write in a clear way the objectives of the work at the end of the Introduction section. We have added a clear objective at the end of the introduction (see below and in page 3 of the revised manuscript, lines 440-446)

“The principal objective of our work was to study the biomedical potential of A. chilense oleoresin by evaluating its capacity to activate PPARγ and to modulate the metabolic dysfunction induced by obesity. We also characterized some of the lipidic components of the oleoresin such as fatty acid, tocopherol and b-carotene and the antioxidant properties in vitro and in vivo using C.elegans as a model”

Point 3: It is not clear what «Gracilex®» is. It is a patent to produce an oleoresin? A certain mixture of lipids with established chemical composition? Has this oleoresin a chemical defined structure? In line 200 says that fatty acids were determined, so the Gracilex has not a chemical defined structure, is it?

To clarify this point we have added the following explanation in the Introduction;  “The oleoresin was produced after organic solvent extraction of dried algae following the procedure published in the patent application WO/2014/186913 and its trademark registered name is Gracilex® “ (see page 3, lines 442-444 in the revised manuscript).  The oleoresin hasn´t have clear chemical defined structure yet. It is a mix of lipids, and from all the possibilities we have characterized the fatty acid profile and the concentration of tocopherols and ß-carotene.

Point 4: Line 199: the name in italic.

The name of A.chilense is in italic (see line 479 in the revised manuscript).

Point 5: it should be advisable to describe the number of the ATTC cell culture lines. 

The ATTC cell culture numbers were included as follows. “HeLa (ATCC® CCL-2), PC12 (ATCC® CRL-1721) and 3T3-L1 (ATCC® CL-173) cell lines (ATCC; Manassas, VA, United States) were used for our studies” (see lines 576-578 in revised manuscript).

Point 6: Conclusions can be improved, more conclusions can be withdrawn from so many tests, for sure.

We have improved and extended the conclusion of the manuscript as follows (se lines 1895-1907 in revised manuscript). “Altogether, our results indicate that Gracilex®, an oleoresin produced using organic solvent extraction from the red macroalgae A. chilense, contains natural PPARγ ligands that increase the PPARγ transcriptional activity. Like SPPARMs, Gracilex® did not induce preadipocyte differentiation as the PPARγ receptor full agonist, rosiglitazone. Gracilex® fed HFD mice showed normalization of glucose and insulin parameters altered by the diet without increasing adiponectin. This result was consistent with the idea that the natural lipids present in Gracilex® act as SPPARM-like agonists of PPARγ. The lipid profile of Gracilex® indicated that the oleoresin was rich in arachidonic acid. Thus, PPARγ agonists present in Gracilex® might be eicosanoids derived from this polyunsaturated fatty acid. Gracilex® showed a high concentration of tocopherols and b-carotene, which correlated with antioxidant effect in vivo using a C.elegans model of oxidative stress. Altogether, our results indicate that Gracilex® represents a good source of natural PPARγ ligands and antioxidants possessing a high nutraceutical value to mitigate metabolic disorders”.

Reviewer 2 Report

Overall, to be frank, it is difficult to understand the paper because of the many undefined abbreviations and inconsistency of the terminologies.

Although it may not change the final output, the outliers of the data should be properly treated (Fig. 1C).

To confirm the antioxidative activity analysis, please try the scavenging effect on DPPH radicals or ABTS.+ assay. The antioxidative assay using nematodes in Figure 5 is not very convincing.

The Supplementary Figure 2 shows body weight changes in the period of one month. The change is only 2 g maximum, which is quite not normal. If the mice were maintained in a not-healthy condition, the nutraceutical effects of  Gracilex® might not be properly addressed.

Minor

Line 2-5:

  • agarophyton” should be written as Agarophyton (Genus name).
  • “pparγ” is an abbreviation of peroxisome proliferator-activated receptor-γ, so it should be abbreviated in upper cases (PPAR-γ).
  • “c57bl/6j” to C57BL/6J
  • caenorhabditis” to Caenorhabditis

Line 34, 197:   use the full name of CUPRAC

Line 187: The term lyophilized is used when the sample was frozen and dried under vacuum. Is this the right condition for drying the sample?

Line 188, 190, 248, 373, 381: Consistency in naming

  • Because of the inconsistent naming of A.chilense oleoresin (Gracilex®), which is the final material extracted from Agarophyton chilense and used in the entire experiments, it is confusing whether the name is the final material or the source of it.

Line 221 and 225: Please clarify what type of the plate was used for the transfection?

Line 333, 437: typos

Line 334: italicize Escherichia coli

Author Response

Answer to Reviewer 2.

Point 1: Overall, to be frank, it is difficult to understand the paper because of the many undefined abbreviations and inconsistency of the terminologies.

We have revised the full manuscript and double checked for consistency in how to name the oleoresin derived from A.chilense. For example, we shortened the introduction, and in the introduction, we added a paragraph where we state the principal objectives of the research and explained how we have named the oleoresin all over the manuscript se below and lines 438-444 in revised manuscript.

“The principal objective of our work was to study the biomedical potential of A. chilense oleoresin by evaluating its capacity to activate PPARγ and to modulate the metabolic dysfunction induced by obesity. We also characterized some of the lipids contained in the oleoresin such as fatty acids, tocopherols and b-carotene and the antioxidant properties in vitro and in vivo using C. elegans as a model. The oleoresin was produced after organic solvent extraction of dried algae following the procedure published in the patent application WO/2014/186913 and its trademark registered name is Gracilex®. From then on, we used always the term Gracilex to define the oleoresin derived from A.chilense red macroalgae along the manuscript. Also, when possible, we have avoided the used of abbreviation and double check that all abbreviations meaning are included in the manuscript.

Point 2: Although it may not change the final output, the outliers of the data should be properly treated (Fig. 1C).

We have analyzed the data of Figure 1C using the GraphPad Software and the Grubbs method for detecting outliers. We just detected one outlier, in the experimental point Gracilex® + T007. Accordingly, we have removed the point from the graph. We did not detect other outliers in the rest of the figures. We have stated this analysis in the section 2.7 Statistical Analysis (lines 834-837 in revised manuscripts).

Point 3: To confirm the antioxidative activity analysis, please try the scavenging effect on DPPH radicals or ABTS.+assay. The antioxidative assay using nematodes in Figure 5 is not very convincing.

There are several reasons why we have chosen the CUPRAC (CUPric Reducing Antioxidant Capacity) method to measure the total antioxidant capacity (TAC) of Gracilex®. This essay is one of the methods used to measure TAC using electron transfer as DPPH radicals or ABTS.+ assays. The advantage of CUPRAC relays over other TAC methods is that they can be used with different matrixes. The CUPRAC method is suitable to measure TAC in edible and essential oils (1,2) and is especially good for samples that required suspension with organic solvents, like Gracilex®(3).

Additionally, it is a simple analytical procedure that does not require expensive equipment and has short reaction times (ref 4). Compared to iron-based antioxidant assays, this method can detect all classes of antioxidants, including thiols, with marginal radical interference (https://www.cellbiolabs.com/sites/default/files/STA-360-total-antioxidant-capacity-assay-kit.pdf). The strategies suggested by Reviewer, DPPH, and ABTS are highly used in the food industry to estimate antioxidants' ability to scavenge free radicals. However, the comparison between the three methods in the literature has been focused mainly on polyphenols and flavonoid antioxidant molecules, the more polar components of oily samples, which are unlikely found in our extract due to the extraction process we have applied. DPPH is mainly used to determine antioxidant potency in more hydrophilic extractions of oils easily diluted in methanol. This is not our case since Gracilex® needs to be suspended in DMSO (5). Similar difficulties with organic solvents are found in the ABTS assay, where ethanol is the solvent of choice. However, oil solutions higher than 1% in ethanol form turbid solution, thus generating problems with measurements (6). Additional to these difficulties, it has been reported that carotenoids interfere in the 515 nm-absorbance measurements of DPPH (7). Taking all these aspects into account, we found that the kit provided by cell Biolabs was a good approach to measure TAC in Gracilex®. The uric acid equivalent has been used in the literature elsewhere and is equivalent to the Trolox equivalents used in other assays (8).

Regarding the suggestion to include another method to determine TAC (such as DPPH or ABTS+) in addition to CUPRAC, the current literature indicates that the antioxidant activity of extracts measured by different assays gives different results depending on the nature of the functional groups responsible for the antioxidant activity (9). Thus, to confirm the antioxidant capacity of Gracilex®, we preferred to perform an in vivo test over another in vitro TAC assay. C. elegans is a well-studied model for testing oxidative stress resistance (10) and, our group has experience working with this model (11,12). Although the Gracilex® effect in this in vivo model could be considered discrete, the data showed a beneficial global effect over survival comparable in magnitude to the well-known pure antioxidant N-Acetyl cysteine.

1.- Özyürek, M. et al. (2011) A comprehensive review of CUPRAC methodology. Anal. Methods 3, 2439-2453. doi: 10.1039/c1ay05320e

2.- Dionysios C. et al. (2014) Evaluation of total reducing power of edible oils. Talanta 130, 233–240.  doi:10.1016/j.talanta.2014.06.058

3.- Apak,R. et al. (2004) Novel total antioxidant capacity index for dietary polyphenols and vitamins C and E, using their cupric ion reducing capability in the presence of neocuproine: CUPRAC method. J. Agric. FoodChem. 52, 7970–7981. doi: 10.1021/jf048741x.

4.- Özyürek, M. et al.  (2011) The main and modified CUPRAC methods of antioxidant measurement. Trends Anal. Chem. 30, 652–664. DOI: 10.1016/j.trac.2010.11.016

5.- Espin JC. et al.  (2000) Characterization of the total free radical scavenger capacity of vegetable oils and oil fractions using 2,2-diphenyl-1-picrylhydrazyl radical. J Agric Food Chem 48: 648–656. DOI: 10.1021/jf9908188

  1. Pellegrini N, et al. (2001) Direct analysis of total antioxidant activity of olive oil and studies on the influence of heating. J Agric Food Chem 49:2532–2538. DOI:10.1021/jf025932w

7.- Noruma, T. et al.  (1997) Proton-Donative Antioxidant Activity of Fucoxanthin with 1,1-diphenyl-2-picrylhydrazyl (DPPH). Biochem. Mol. Biol. Int., 42, 361-370. DOI: 10.1080/15216549700202761

8.- Agar, O. T. et al.  (2015). Comparative studies on phenolic composition, antioxidant, wound healing and cytotoxic activities of selected Achillea L. species growing in Turkey. Molecules, 20(10), 17976-18000. Doi: 10.3390/molecules201017976

9.- Sethi, S. et al.  (2020) Significance of FRAP, DPPH, and CUPRAC assays for antioxidant activity determination in apple fruit extracts. European Food Research and Technology 246:591–598 DOI:10.1007/s00217-020-03432-z

10.- Ayuda-Durán, B. et al.  (2020) Caenorhabditis elegans as a Model Organism to Evaluate the Antioxidant Effects of Phytochemicals. Molecules. Jul; 25(14): 3194. doi: 10.3390/molecules25143194

11.- Minniti A.  et al (2009) Methionine sulfoxide reductase A expression is regulated by the DAF-16/FOXO pathway in Caenorhabditis elegans. Aging Cell, Dec;8(6):690-705.  doi: 10.1111/j.1474-9726.2009.00521.x.

Point 4: The Supplementary Figure 2 shows body weight changes in the period of one month. The change is only 2 g maximum, which is quite not normal. If the mice were maintained in a not-healthy condition, the nutraceutical effects of Gracilex® might not be properly addressed.

The reviewer is right, according to the weight curves of JAX® mice they should have gain 4 grams in one month. The difference between these mice and the mice used in HFD experiments is that these mice were fed normal chow diet (the one bought by the veterinarians in the animal facility) and also, they were from a different colony (the colony bred in the animal facility of the Catholic University corresponding to C57BL/6J). It is possible that the normal chow diet is lower in calories compared to the chow diet used in The Jacksons Laboratory. This experiment was specifically done to test the toxicology of Gracilex® and to check the effect in weight gain that was not different than control. The mice look healthy and mated normally under all inspection made by veterinarians in the animal facility.

The mice used to investigate the effect of Gracilex® in HFD-fed animals were directly bought to The Jackson Laboratory and fed with LFD and HFD food bought from Research Diet, Inc for these specific experiments.

Minor Point 1: In the title; “agarophyton” should be written as Agarophyton (Genus name).“pparγ” is an abbreviation of peroxisome proliferator-activated receptor-γ, so it should be abbreviated in upper cases (PPAR-γ).“c57bl/6j” to C57BL/6J “caenorhabditis” to Caenorhabditis. The Capitals were rectified in the title. Lines 1-5 in revised manuscript.

Minor Point 2: use the full name of CUPRAC. We have used the full name of CUPRAC when presenting the method in the section 2.3.2. as follow. “Determination of total antioxidant capacity of Gracilex® using a CUPRAC (CUpric ion Reducing Antioxidant Capacity) assay (line 556 in revised manuscript).

Minor Point 3: Line 187: The term lyophilized is used when the sample was frozen and dried under vacuum. Is this the right condition for drying the sample? Yes, this is the right condition for drying the sample. The fallowing paragraph was changed/added to the section 2.3.1. A.chilense (Gracilex®) oleoresin production (line 447 in revised manuscript). “Chopped alga was freeze-dried through lyophilization. Then, to produce a lyophilized powder the dried material was grinded using a coffee grinding machine. Dichloromethane was used as an organic solvent for lipid extraction of pulverized dried seaweed”.

Minor Point 4: Line 188, 190, 248, 373, 381: Consistency in naming. Because of the inconsistent naming of A.chilense oleoresin (Gracilex®), which is the final material extracted from Agarophyton chilense and used in the entire experiments, it is confusing whether the name is the final material or the source of it. We agreed with the reviewer as indicated in the response to Point 1 we have explained what Gracilex® is in the introduction and from there on we have used just Gracilex® to indicate the oleoresin.

Minor Point 5: Line 221 and 225: Please clarify what type of the plate was used for the transfection?

PPAR reporter activity was performed in HeLa and PC12 cells seeded in 24 well plates. PPARγGAL4 transactivation assays were performed in PC12 cells seeded in 48 well plates.   This is indicated in the section 2.4.2. Cellular transfection. The Hela cells was erased from the PPARγGAL4 transactivation assays that were just performed in PC12 cells.

Minor Point 6: Line 333, 437: typos Line 334: italicize Escherichia coli

Formed lines 333 and 334 now reads “2.6. C. elegans studies. Nematodes were cultivated at 20ºC under standard laboratory conditions on agar plates fed with Escherichia coli (OP50) as a food source (58)”. Lines 797-799 in revised manuscript.

Typo in former line 437, second “with” was changed to “without” in line 1064 in revised manuscript, (second last line of Fig.1 legend)

Round 2

Reviewer 2 Report

Comment on Fig 1:

The switching of the promoter region from PPRE to Gal4 MH100 and swapping of the DNA binding domain of PPAR-r dramatically increased the activity of RGD at 1uM concentration from 4-5 fold to over 120-fold. However, the treatment of Gracilex did not change the relative PPAR-r activity. This might suggest that Gracilex may not be an efficient PPAR-r activator. How can the marginal activation of PPAR-r by Gracilex trigger the relevant biological activities? (For example, if you treat an HFD-fed mouse model with 40 nM of RGD, can you still expect the PPAR-r activation?)

In Fig 1C, 

The antagonist, T0070907, at 10 uM concentration could very efficiently block the activity of Gracilex. This result again suggests that Gracilex is a marginal PPAR-r activator with a low affinity to the receptor.

With those questions on Figure 1, I would raise questions to the authors to reconcile the marginal PPAR-r activation by the Gracilex treatment and the insulin-sensitizing effects.

Comment on Fig 5:

The inconsistent time-to-%-active nematodes curves in the control groups (eg, less than 40% in B but over 50% in C at 80 min) make reliable comparisons with the treated group (Gracilex or NAC) difficult.

Is it possible to perform the experiments in Fig. 5B and C simultaneously and put the results in one figure? When you do the experiment, please include tocopherol and b-carotene control using a similar amount containing in the Gracilex treatment.

Author Response

Round 2. Reviewer 2 comments.

Reviewer comment: English language and style are fine/minor spell check.

Response: We have submitted the paper to professional English editing using the MDPI system

Reviewer comment: The switching of the promoter region from PPRE to Gal4 MH100 and swapping of the DNA binding domain of PPAR-r dramatically increased the activity of RGD at 1uM concentration from 4-5 fold to over 120-fold. However, the treatment of Gracilex did not change the relative PPAR-r activity. This might suggest that Gracilex may not be an efficient PPAR-r activator. How can the marginal activation of PPAR-r by Gracilex trigger the relevant biological activities? (For example, if you treat an HFD-fed mouse model with 40 nM of RGD, can you still expect the PPAR-r activation?)

In Fig 1C, 

The antagonist, T0070907, at 10 uM concentration could very efficiently block the activity of Gracilex. This result again suggests that Gracilex is a marginal PPAR-r activator with a low affinity to the receptor.

Response: The reviewer's observations regarding the PPARg activities induced by the different agonists in the two transactivation assays are undoubtedly valid. However, we kindly disagree with the idea of comparing both transactivation assays directly and with the statement that the activity we are measuring is marginal. 

First, to study whether Gracilex@ contains natural agonists of PPARg we have used two different transactivation assays based on the transfection of cell lines with two plasmids, as mentioned by the reviewer. We started our studies by using an assay based on the PPRE response element, which has been extensively used to study PPAR agonists in a cell context.  The plasmid-encoded the full PPARg  receptor and the second plasmid contained a PPRE response element upstream of a reporter gene (luciferase). In this assay, although we overexpressed PPARg, the response will depend on the heterodimerization with the retinoic acid receptor (RXR) and the expression of the endogenous PPAR receptors that may interfere with the assay since all PPARs will interact with the PPRE response element (common for all 3 PPARs alpha, beta/delta, gamma). In this assay, we compared Gracilex@ with rosiglitazone (1 uM). This concentration of rosiglitazone is known to induce the maximum response in this type of assay. Commonly, the types of agonists are defined by comparing them to a full agonist of the receptor such as, rosiglitazone. In this assay, Gracilex lipids accounted for up to 60% of PPARg activity induced by rosiglitazone (Figure 1B). This result, classify the Gracilex® natural lipids as partial agonists of the receptor, which is not surprising since most of the natural ligands found for PPARg are partial agonist and therefore weaker than rosiglitazone (1).  The second assay we used consisted of the transfection of two plasmids, one encoding a chimeric protein containing the ligand-binding site of PPARg  and the DNA binding domain of Gal4, the second plasmid had a Gal4 response element upstream of a reporter gene (also luciferase). This second assay is more specific than the first one since luciferase activity is a direct consequence of agonist binding to the LBD- PPARg  and there is no background noise created by the fact that the full PPARg  required heterodimerization with RXR or the presence of endogenous PPARs. Therefore, both assays are not comparable. Furthermore, as we already knew that Gracilex® contained partial agonists, we compared Gracilex® activity with the activity induced by two different and well described partial agonist of PPARg that is FMOC-Leu and INT131 at a high concentration that is 25 times higher than for rosiglitazone (25 uM) known to induce their maximum response. When comparing with FMOC-Leu and INT131, we found that Gracilex® activity was comparable with them, reaching the same levels of luciferase induction (Figure 1B). 

To the question broad by the reviewer, how can the marginal activation of PPARg  by Gracilex trigger the relevant biological activities? Some partial agonists are just weaker agonists than full agonists as rosiglitazone, but others are classified as selective PPAR modulators or SPPARMs, meaning that the doses/response curve is uncoupled to the efficacy of difference responses triggered by full agonists (2,3). For example, INT131 can induce insulin sensitizer effects with the same efficacy compared to rosiglitazone without increasing the adipose tissue in T2DM murine models (4,5) or induce adipocyte differentiation in vitro as shown by others and by us for INT131 and Gracilex (Figure 2)(3). This is because PPARg  has the most oversized binding pocket among the nuclear receptors, allowing entry and binding of up to two different molecules in a covalent or non-covalent manner, inducing different conformation of the receptor. This is transduced in differential recruitment of co-activators and hence, transcriptional responses (3,4,5). The fact that T0070907, at 10 uM concentration can efficiently block the activity of Gracilex supports the idea that the assay used is very specific and has biological meaning.

Reviewer comment: Comment on Fig 5:

The inconsistent time-to-%-active nematodes curves in the control groups (eg, less than 40% in B but over 50% in C at 80 min) make reliable comparisons with the treated group (Gracilex or NAC) difficult.

Is it possible to perform the experiments in Fig. 5B and C simultaneously and put the results in one figure? When you do the experiment, please include tocopherol and b-carotene control using a similar amount containing in the Gracilex treatment.

Considering the reviewer observation, a new graph was incorporated to compare all the independent assays at their maximum protection values (Figure 5D). The data compared the maximal effect observed between 70 to 90 minutes after the start of the challenge. In each assay, the percentage of active individuals was normalized against its control. It is important to consider that the assay was performed using at least 30 individuals per condition in 3 different assays. We found statistical differences between the response curves when compare treated individuals with control. Nevertheless, these responses show variation in the time of the maximal effect of the antioxidants between 70 min to 90 min for NAC and 80 min for Gracilex®.

Regarding the point of including tocopherol and b-carotene as controls, we acknowledge the contribution of using both molecules to understand the molecular mechanism responsible for Gracilex® antioxidant capacity. This critical question will be covered in a future study. We chose N-Acetyl-L-Cysteine (NAC) antioxidant molecule to compare in vivo antioxidant action because it is a well-known antioxidant commonly used C. elegans studies. Besides, our group already tested the beneficial concentration of NAC for the restoration of redox balance in the msra1 mutants (6). It is well known that the organism's response to antioxidants displays hormesis, since low levels of these compounds may be beneficial while higher concentrations tend to be detrimental (7). The use of other antioxidants requires the adequate standardization as a positive control for our assays or for characterizing the active component of Gracilex®.

We have added the following indication in page 17 lines 2334-2337 in the revised text “Similar behavior was observed in the nematodes previously exposed to NAC, a well-known potent antioxidant which have been demonstrated that rescues redox homeostasis in msra-1 mutant a dose of 5 mM (59). The time response curve in Figure 5B show the highest protection of Gracilex® at 80 minutes. This time response is in the range of NAC between 70 to 90 min after peroxide challenge (Figure 5C and D)”.

  1. Wang, S., Dougherty, E. J., and Danner, R. L. (2016) PPARγ signaling and emerging opportunities for improved therapeutics. Pharmacological Research 111, 76-85
  2. Higgins, L. S., and Depaoli, A. M. (2010) Selective peroxisome proliferator-activated receptor gamma (PPARgamma) modulation as a strategy for safer therapeutic PPARgamma activation. The American journal of clinical nutrition 91, 267S-272S
  3. Rocchi, S., Picard, F., Vamecq, J., Gelman, L., Potier, N., Zeyer, D., Dubuquoy, L., Bac, P., Champy, M.-F., Plunket, K. D., Leesnitzer, L. M., Blanchard, S. G., Desreumaux, P., Moras, D., Renaud, J.-P., and Auwerx, J. (2001) A Unique PPARγ Ligand with Potent Insulin-Sensitizing yet Weak Adipogenic Activity. Molecular Cell 8, 737-747
  4. Dunn, F. L., Higgins, L. S., Fredrickson, J., and DePaoli, A. M. (2011) Selective modulation of PPARγ activity can lower plasma glucose without typical thiazolidinedione side-effects in patients with Type 2 diabetes. J Diabetes Complications 25, 151-158
  5. Lee, D. H., Huang, H., Choi, K., Mantzoros, C., and Kim, Y. B. (2012) Selective PPARgamma modulator INT131 normalizes insulin signaling defects and improves bone mass in diet-induced obese mice. Am J Physiol Endocrinol Metab 302, E552-560
  6. Minniti, A. N., Arriagada, H., Zúñiga, S., Bravo-Zehnder, M., Alfaro, I. E., and Aldunate, R. (2019) Temporal pattern of neuronal insulin release during Caenorhabditis elegans aging: Role of redox homeostasis. Aging Cell 18, e12855
  7. Schulz, T. J., Zarse, K., Voigt, A., Urban, N., Birringer, M., and Ristow, M. (2007) Glucose restriction extends Caenorhabditis elegans life span by inducing mitochondrial respiration and increasing oxidative stress. Cell Metab 6, 280-293
